# Social Support Initiatives That Facilitate Exercise Participation in Community Gyms for People with Disability: A Scoping Review

**DOI:** 10.3390/ijerph20010699

**Published:** 2022-12-30

**Authors:** Rachel A. Kennedy, Georgia McKenzie, Carlee Holmes, Nora Shields

**Affiliations:** 1Department of Physiotherapy, Podiatry and Prosthetics and Orthotics, La Trobe University, Bundoora 3086, Australia; 2CP Achieve Centre of Research Excellence, Neurodisability and Rehabilitation, Murdoch Children’s Research Institute, Parkville 3052, Australia

**Keywords:** physical activity, fitness, adolescents, young adults, supervision, recreation centre, peer support

## Abstract

People with disability report social support facilitates participation in physical activity. A scoping review explored social support strategies used to facilitate exercise participation for people with disability (aged ≥ 15 years) in community gym settings. Seven electronic databases were searched. Studies were screened for eligibility based on title and abstract followed by full-text review. Data were analysed using content analysis and narrative synthesis. Forty-two articles reporting data from 35 studies were included. Eight types of social support were identified: supervision (*n* = 30), peer support (*n* = 21), specialist support (*n* = 19), orientation (*n* = 15), education (*n* = 7), logistical support (*n* = 6), motivational support (*n* = 5) and organised social activities (*n* = 4). Direct supervision was typically provided 1:1 or in small groups by staff experienced working with people with disability. Peer support typically involved support from exercise group participants or a peer mentor. Specialist support was usually provided by a health or exercise professional either directly to people with disability or to the people providing support to them (e.g., trainer). Orientation to the gym environment, equipment and exercise program was usually provided over 1 or 2 sessions. Gym staff may use these strategies to guide the implementation of social supports within their facilities to promote social connectedness and participation for people with disability.

## 1. Introduction

Participation in physical activity among adolescents and adults with disability is primarily influenced by environmental factors, particularly social support [1,2,3]. Social support relates to formal or informal understanding (emotional support), tangible assistance (instrumental support), advice (informational support) or feedback (appraisal support) [4] that contributes to the capability, opportunity and motivation of people with disability to be active [3]. Empirical studies consistently report the importance of social support as an enabler of physical activity in children, young people, and older adults with disability [1,2,5]. Social support is described as a positive influence on the beliefs, experiences and prioritisation of physical activity for people with disability, and to their sense of self [3]. Social support can be provided by anyone within a social network (e.g., family, friends, peers, staff, professionals, organisations, policymakers) however for many people with disability, social support to participate in physical activity across their lifespan is often lacking within community settings or provided primarily by their families [3].

Article 30 of the United Nations Convention on the Rights of Persons with Disabilities affirms the obligation to support the participation of young people with disability in recreational and sporting activities [6]. Facilitating participation in physical activity is a priority for adolescents and adults with disability. Most do not participate in the recommended amounts of activity [7] and many are unable to access physical recreation opportunities available within their communities. Access to community-based activities and facilities is particularly important during the transition from adolescence to young adulthood. This period coincides with a sharp decline in physical activity, reduced social participation, increased social isolation, heightened psychological vulnerability [8] and reduced access to support services [9,10]. Physical activity confers physical, mental and social health benefits and people with disability who are physically active report increased social interactions, feel socially connected and have a greater sense of independence and confidence [11]. Health behaviour change theories explain the relationship between social support and physical activity participation. Self-determination theory [12] and social cognitive theory [13] propose that health behaviours can be fostered through social interactions (‘relatedness’) and learning from others (‘modelling’), respectively. The availability of social support, together with positive social connectedness, and a suitable physical environment are essential to being physically active for people with disability [3].

Gyms are a preferred [14] and socially meaningful setting for physical activity for people with disability. A lack of social support acts as a barrier to participation in community gym settings [5], while having someone to exercise with is a key enabler [5,15]. Research shows gyms can be an intimidating space for people with disability, who describe fears of ‘standing out’, self-consciousness and ‘not-belonging’ [15,16]. Having social support in the gym can facilitate participation via provision of emotional support (e.g., enable a sense of belonging [17,18]), instrumental support (e.g., assistance navigating basic exercises, using equipment or staying focused [5]) and informational support and appraisal (e.g., education on healthy lifestyle choices and motivational feedback). There are many ways in which community gym and recreation facilities could implement social support to facilitate attendance and positive physical activity experiences. The aim of this study was to describe how social support has been implemented for people with disabilities in research studies conducted in community gyms.

## 2. Methods

### 2.1. Protocol and Registration

A scoping review was completed in accordance with recommendations from the Joanna Briggs Institute [19] and reported according to the PRISMA Extension for Scoping Reviews (PRISMA-ScR) checklist (Appendix B) [20]. Scoping reviews provide an overview of the evidence, systematically mapping the available literature on a topic [21] and examining the extent (size), range (variety) and nature (characteristics) of the evidence [20]. They also enable a broad examination of available evidence to identify knowledge gaps, to clarify key concepts related to the area of interest and to report the types of evidence that may inform health practice and practice change [22]. A review protocol was published prospectively on the Open Science Framework [22].

### 2.2. Eligibility Criteria

Reports of primary studies (any design) of physical activity interventions were included if they were completed in a community gym or recreation centre and used social support strategies, either alone or as an adjunct, to facilitate the participation for adolescents and adults (aged ≥15 years) with disability. For the purposes of the review, disability included childhood-onset disabilities (for example, cerebral palsy or Down syndrome), disabilities occurring in childhood or adulthood (for example, stroke or acquired or traumatic brain injury) and adult-onset conditions (for example, multiple sclerosis or Parkinson’s disease). Disabilities due to primary musculoskeletal disorders (for example, low back pain or osteoarthritis), pain and fatigue disorders, fragility, ageing (including dementia) and acute psychiatric conditions were excluded. Only full text, peer-reviewed, English-language articles were included. Systematic or narrative reviews, protocol papers, conference abstracts and theses were excluded.

A community gym or recreation centre was defined as a publicly owned and funded facility operated by either a local (municipal) government authority or by a management company on that authority’s behalf. Physical activity interventions needed to have been completed in a gym setting open to the public (including those operating on university or college campuses) that included exercise equipment (for example, stationary bikes or pin-loaded weights machines) with or without a functional exercise space. Programs that took place in a group exercise space (for example, a yoga studio) that was segregated from other public users of the gym or in a research laboratory were excluded.

### 2.3. Search Strategy

A systematic search (Appendix C) was completed of the following databases: CINAHL, Medline, EMBASE, PsychINFO, SPORTDiscus, PubMed (last 12 months only) and PEDro. The search strategy included keywords and MeSH headings related to the concepts of ‘disability’, ‘physical activity’ and ‘community gym’. Database searches were supplemented by citation tracking using Google Scholar and by reviewing the reference lists of included studies.

### 2.4. Selection of Sources of Evidence

Search yields were imported into Endnote (version 20), duplicates deleted and then exported to Covidence. Titles and abstracts were assessed against the eligibility criteria by two reviewers (RK and NS), independently. Where eligibility could not be determined from the title and abstract alone, the full text of an article was retrieved and examined before a final decision on eligibility was made. Full text articles were examined by two reviewers (RK and CH or GMc) and any disagreements were resolved by consensus. The authors of full text articles were contacted via email to clarify details about their studies, where necessary.

### 2.5. Data Charting Process and Data Items

Data from included articles were extracted using a data extraction sheet developed for the review in Microsoft Excel. Data were extracted by one reviewer (RK) and checked by a second reviewer (NS). Any disagreements were resolved by consensus. The following data were extracted: bibliographic information (author(s), year of publication, title, country of origin), study aim, study characteristics (sample size, participant demographics, recruitment setting, study design and/or methods), details of the physical activity intervention (including duration, setting, frequency), description of the social support strategy(ies) (including what support, the purpose, who provided, training received, and how implemented). Clinical outcomes from the included studies were not reported as social support is usually implemented as part of a complex intervention making it difficult to conclude that specific outcomes resulted from social support initiatives alone. Study quality was not formally appraised as the aim of the review was to describe how social support had been implemented.

### 2.6. Data Synthesis

Descriptive statistics summarised the characteristics of the included studies. Content analysis including the types and sub-types of social supports and narrative synthesis were completed to describe how social support had been implemented in community gyms. Data were managed in Microsoft Excel. First, each paper was systematically coded to identify the social support strategies implemented. Social support strategies were then grouped into themes based on similarity of purpose. Descriptive labels summarising the purpose of the supports were applied to each theme and categorised according to Berkman [4]. Lastly, each social support was mapped to the socio-ecological model. At least two authors conferred on each stage and cross-checked theme groupings and categorisation.

## 3. Results

The search strategy identified 1517 articles for screening, with 111 articles undergoing full text review. Forty-two articles reporting data from 35 studies published between January 2004 and July 2022 were included (Figure 1).

### 3.1. Characteristics of Included Studies

The studies included various congenital (for example, Down syndrome and cerebral palsy) and adult-onset disabilities (for example, multiple sclerosis and Parkinson disease) with between 3 and 163 participants of mean age from 15 and 70 years (Table 1). Studies included randomised controlled trials (*n* = 21) [14,17,18,23,24,25,26,27,28,29,30,31,32,33,34,35,36,37,38,39,40]; pre-test post-test studies (*n* = 9) [41,42,43,44,45,46,47,48,49] three of which included qualitative sub-studies [15,50,51]; feasibility studies (*n* = 2) [52,53], a case report [54], an observational study [55] and a single-subject design [56]. Studies were conducted in developed countries, with one exception [48] (Table 1). The physical activity interventions implemented included strength, aerobic or balance training, or a combination of these (Table 2). The duration of the interventions was 3 weeks to 24 months, with most programs running for 10 to 12 weeks.

### 3.2. Social Support Strategies

Social supports used in the included studies were grouped into categories according to Berkman et al. [4] (Appendix D).

#### 3.2.1. Supervision

Direct supervision was support provided for the duration of each exercise session, usually for the purpose of providing instrumental, informational and appraisal support, and in some instances emotional support. Supervision was provided in 30 of the 35 included studies, usually by gym staff or by allied health professionals (for example, physiotherapists) however in five studies, supervision was provided by volunteers [38,44,49,54] or paid support workers [45] (Appendix A). The supervisor to participant ratio was typically high; either 1:1 or small groups (i.e., ratio of 1:2, 1:3 or 1:4) although larger groups of 10 to 12 participants per session did occur [41,42]. Direct supervision was usually provided for the duration of the intervention, except in one study [52] where it was provided during the first exercise session only, after which participants with glioma exercised in the gym unsupervised (A. Hansen, personal communication, 10 May 2022).

Typically, direct supervision was provided by staff experienced in working with people with disability or by staff or students who had received training. Examples of the type of training completed by those providing supervision were 2- or 3 h formal education sessions [41,44,49,55,57], receipt of a training manual [14,32], attending sessions with a physiotherapist or exercise specialist [24,26,30,48,54] or a combination of these [27,37,40,56]. The content of formal training sessions included information about disability, communication strategies, managing challenging behaviour, social interaction strategies, exercise prescription and progression, motivational techniques and providing feedback.

#### 3.2.2. Peer Support

Peer support was included in 21 of 35 studies (Table 2) and was primarily providing emotional and instrumental support. This type of support was provided by disabled peers in supervised group settings, or by non-disabled peers (mentors, usually volunteer or paid university students) who provided 1:1 support (Appendix A). Group exercise programs used three formats: small groups of 2 to 4 participants [31,33,35,36,45,47,51,55,57,58], large groups of 10 to 15 participants [29,38,39,44,49] or a large group divided into smaller sub-groups (for example, 12 participants exercising in smaller groups of 3 or 4) [25]. During group exercise, participants either completed an individually tailored program alongside other participants [41] or participants with a similar level of independence exercised together [42].

One-to-one peer support was provided by a university student in seven studies [17,18,29,34,44,53,54]. In four of these seven studies [17,18,34,53], student mentors exercised alongside the participant with disability, unless the participant had a complex disability (usually a young person with a significant intellectual disability and/or communication impairment and/or epilepsy) [34]. In one study, participants were assigned a volunteer (university student) exercise partner during the transition (over 5 to 9 weeks) from supervised exercise in a small group area to independent exercise in the main gym [44]. Peer support was provided by high school students in two studies, who exercised in pairs (1:1) or triads (1:2) with participants with disability, as part of a larger supervised exercise program [55,57]. The roles and responsibilities of mentors included counting repetitions, motivating and providing feedback, maintaining exercise logbooks, problem solving and adapting exercises and equipment in consultation with specialists such as a physiotherapist, assisting to get on/off equipment and ensuring safe technique and equipment use.

#### 3.2.3. Specialist Support

Specialist support was usually provided by a health professional (for example, a physiotherapist) or an exercise professional (for example, an exercise physiologist) (Appendix A) and covered all four domains of support, that is, emotional, instrumental, appraisal and informational support. Specialist support was provided to participants with disability [23,28,29,52] or to people who provided peer support or supervision (for example fitness trainers and student mentors) [14,17,18,27,31,53,56,59] or both [26,34,37,41,44,48,60]. The type of specialist support provided included in-person gym visits [23,26,31,34,37,41,48,56,59,60], 2 to 3 h training sessions for gym staff or student mentors [14,17,18,26,27,34,41,53,56,59,60] and remote support via telephone, email or an online exercise App [14,18,23,32,34,37,49,52]. In-person visits included orientation sessions or providing gym staff and student mentors with specialist advice on motivational and behavioural management strategies [17,18,34,37,53] or adapting equipment and exercises [34,44] for participants with disability. The amount of specialist support varied and included regular scheduled support or support provided only as required [14,17,18,23,34,52,53,60]. For example, in one study, for a subset of participants with complex disability, a physiotherapist attended their first gym session and provided weekly remote monitoring thereafter for the duration of the program [34].

#### 3.2.4. Orientation and Familiarisation Sessions

Orientation sessions for the person with disability was included in 15 of 35 studies (Appendix A) to provide informational support. Typically, this comprised one or two sessions although up to 15 sessions [56] were reported. The intention of orientation was to familiarise the person with disability to the gym environment and to gym equipment, to ensure safety, and to make any necessary adaptations to the exercise program. Participants completed these sessions with a physiotherapist (*n* = 4) [26,41,43,48] or gym instructor (*n* = 6) [23,27,34,45,47,52], both a physiotherapist and a gym instructor (*n* = 1) [51] or a member of the research team (*n* = 4) [49,54,56,59].

#### 3.2.5. Other Social Support Strategies

Additional types of social supports reported less commonly were ‘education’, ‘organised social activities’, ‘motivational strategies’, and ‘logistical support’.

Formal education provided informational support and was delivered to participants with disability in seven studies by staff with expertise in exercise, disability, nutrition and/or health education [26,28,29,30,49,56,59] (Appendix A). Education was delivered using written materials, video assisted schedules delivered via electronic tablet, workshops, 1:1 consultation with participants and in-person during exercise sessions. Education included information about initiating and maintaining safe exercise and physical activity participation in the gym and healthy lifestyle choices.

Motivational strategies encompassed appraisal and emotional support and included goal setting [14,59], motivational interviewing [30], “checking in” if exercise sessions were missed [24], the use of a web-based application with the capacity to share via social media [30,61] and support from coaches of an external organisation (Special Olympics) [29].

Organised social activities were used to provide emotional support and were included in four studies of adult participants only [25,46,49,50]. These activities included socialising with other participants from the program following exercise sessions for light refreshments [25,46,50]. One study [49] provided opportunities for participants to meet at a local park for recreational sports and at a restaurant for meals during the study.

Transport to and from the gym was the most common type of logistical support provided and included instrumental and practical supports such as transport by shuttle bus, taxi or organised transport by ancillary programs attended by participants (for example, day programs) [33,35,38,54,60]. In one study, gym staff met participants in the car park to assist them into the facility [60].

### 3.3. Mapping Social Supports to the Socio-Ecological Model

These eight strategies map to the interpersonal and organisational levels of the socioecological model, respectively and allow an understanding of the complex interplay and range of social support strategies that young people with disability often need to successfully participate in physical activity within the gym setting (Figure 2). The identified initiatives comprise formal structured supports providing tangible instrumental assistance (supervision, peer support, specialist support and logistical support), emotional understanding (supervision, peer support, specialist support, motivational, organised socialisation), appraisal and decision making (supervision, peer support, specialist support, motivational) and informational support (supervision, specialist support, orientation, education). There were also examples of informal and unstructured supports that are less quantifiable but equally important, for example, the support offered by peers in a group exercise class. A shared element of these supports was people; the three most common social supports were person-based, that is, supervision, peer support and specialist support.

## 4. Discussion

Of the eight identified social supports that have been used in research to facilitate exercise, person-based support appears fundamental to successful implementation and integration of people with disability in community gym settings. The person-based social supports reported most often involved direct supervision, usually by someone with disability experience, for example a gym staff member or health professional (most often a physiotherapist), working either 1:1 or in small groups. This is consistent with qualitative studies where participants with disability reported that supervision from a knowledgeable supervisor or mentor was important to them [34,37,39,41,44,48,50,51,52,53].

One-to-one support is resource intensive and differs from the common practice in gym settings of indirect supervision where a gym trainer oversees all members exercising in the gym space. The provision of social support can facilitate participation in gym settings, but it is important to consider how such support can be implemented outside of funded trials. Disability funding initiatives, such as the National Disability Insurance Scheme (NDIS) in Australia [62], may cover the costs of 1:1 support in the gym, from a support worker, personal trainer or specialist (e.g., physiotherapist or exercise physiologist). Collaboration between gym facilities, local government, funding bodies (e.g., NDIS) and available supports (e.g., specialists and carers) provide an opportunity to maximise the resources and skills of each sector to facilitate the participation of people with disability in community gym settings.

People with disability describe feeling more confident exercising with a support person when that support person has received training and understands disability and their individual needs [34]. Disability specific knowledge and skills of staff are often identified as important factors in facilitating (or not) the participation for people with disability in community gym settings. In this review, when the person providing supervision did not have disability experience, they were often provided with training about disability and supported by a specialist when adapting exercises and equipment or both. The provision of specialist support, either directly to a person with disability or indirectly via support to gym staff is therefore an important consideration for implementation of community-based physical activity programs. Provision of specialist support can bridge the gap between a rehabilitation “therapy-model” of exercise to a community-based physical activity participation model and simultaneously support the upskilling of gym staff. While specialist support within the included studies was most often provided in person, it could also be provided remotely when the infrastructure, organisation-based support and funding is in place to provide it. Providing remote specialist support facilitates access for people with disability living in regional and rural areas where specialist support might not be as readily available.

Peer support, provided by people with and without disability, was usually an adjunct to specialist support and was important to those with disability [34,39,47,50,51,53]. When a non-disabled peer exercised alongside a person with disability, their role was of a peer mentor or exercise partner rather than as an exercise instructor [17,34,53,55,57]. The non-disabled peer mentor model fostered social connection, particularly for adolescents and young adults with disability who appreciated the time taken to build rapport, making exercise more fun and being accountable to an exercise partner [15]. Peer support instils confidence in people with disability to exercise in a community gym setting, ensuring safety and as a source of social connectedness, accountability, and encouragement [3,15]. Similarly, the inclusion of caregivers and partners, usually a spouse, in group programs for older adults provided motivation to “stick” with the program [41]. Skilled strategies (such as motivational interviewing [30] and goal setting [30,40]) and practical motivational strategies (such as checking-in with participants who missed sessions [24]) were also implemented to facilitate adherence and sustained participation in the gym setting. Within the included studies, peer support was primarily facilitated by the research teams. Further implementation research is needed within gym settings to better understand how facilities might support person-based one-to-one social support initiatives as an alternative to standard gym practice.

Orientation and familiarisation were widely utilised in the included studies and mirrors standard practice, making this an appealing strategy for gym facilities given infrastructure is likely to already exist to support implementation. However, it is important the content and delivery of orientation sessions are tailored to the needs of people with disability, including how to exercise with good form, using equipment safely and with correct technique and injury prevention. It is also important to recognise a single orientation session may not be sufficient for a person with disability. In the studies included in this review, between 2 and 15 orientation sessions were implemented [56]. People with disability, particularly those with intellectual disability may need additional support so that information might be repeated, reinforced and possibly delivered in alternative formats at subsequent gym sessions. Other organisational level strategies targeted exercise participation indirectly through education, logistical support and facilitating group cohesion via organised social activities. The latter was only reported in studies involving older adults.

Although resource-intensive, person-based supports appear important in facilitating participation among people with disability in community gyms settings. All the included studies were funded research trials that provided the organisational framework and covered most, if not all, the costs of implementation including salaries, workforce training, educational resources and technology. Outside of research trials, it is unclear how the identified social supports would be implemented sustainably by gym facilities, and how the cost of social supports would be funded. Disability funding initiatives, such as the NDIS in Australia, often pay for some of the costs of community participation, such as the cost of specialists and support staff. However, the cost of entry fees or gym membership is often not covered. In some instances, philanthropic and not-for-profit organisations offer reduced, low or no-cost membership options for disadvantaged groups, including people with disability. These programs, designed to bridge the equity gap and enable access and inclusion for all to gym and recreation facilities are not, however, widespread across the industry. Cost is a known barrier to exercise participation for most people with disability [1], who are often unemployed, under-employed or in low-paying jobs. Given the physical, mental and social connection benefits of physical activity participation for people with disability [63,64], there is a need for economic evaluation from a societal perspective of social support initiatives within gym and recreation facilities to inform policy.

The social supports identified in this review primarily map within the socio-ecological model to person-based supports and organisational-based support for the individual with disability and their close social networks (Figure 2). However, no studies included in this review considered the potential impact of wider organisational, physical and social environmental or policy contexts that could facilitate or hinder the implementation of social support strategies within community gym settings. Local champions may drive practice in the provision of social support for people with disability within individual community facilities. However, the support of decision-makers within recreation organisations and the local government ultimately influences the implementation of inclusive policies and importantly, funding to support these policies. The findings of this review provide a basis for how recreation gym facilities could implement social support to increase participation among people with disability. It also provides exemplars of strategies or programs that could be supported by local government policy and funding.

A strength of this review is its size. Data from 35 exercise trials (reported in 42 articles) involving adolescents and adults with congenital, acquired and progressive disabilities, were synthesised, and eight types of social support strategies were illustrated. These strategies, applied in various forms, have been used by research programs to successfully facilitate the physical activity participation of people with disability in community gym settings. These strategies provide an ‘ideas bank’ that can be used by organisations to guide implementation within their gym facilities to promote social connection and participation of people with disability. There are limitations in this review relating to the facilities, settings and infrastructure to which the findings primarily relate (community gyms). The types of facilities in this review are more likely to be available in developed countries and may be governed by different policies and funding sources compared with privately owned gyms. Local government facilities usually exist to provide access to recreation for the whole community, including those with disability, and therefore implementing social support within these settings would align with health and inclusion policies. This scoping review is the first of a three-phase project investigating how social supports are implemented in current practice in community gyms and the perspectives and lived experience of this type of support among young adults with disability who attend community gyms [22]. The principles underpinning many of the social support strategies identified in this review may also have applications beyond community gym settings.

## 5. Conclusions

This review identified eight social support initiatives that can be implemented in community gym settings to facilitate the participation of people with disability in physical activity and promote social connections within their community more broadly, as part of their everyday lives, like their peers without disabilities [65]. Supervision, peer support, specialist support and orientation were the most utilised strategies. Collaboration and partnerships between researchers, rehabilitation specialists (e.g., physiotherapists), recreational organisations (e.g., gym staff and management) and government and insurance policymakers are important to better understand how these initiatives might be implemented and operationalised in practice. Transitioning exercise from a medical rehabilitation model to a community-based participation model is important in providing access to and facilitating safe, individually tailored physical activity for young people with disability.

## Figures and Tables

**Figure 1 ijerph-20-00699-f001:**
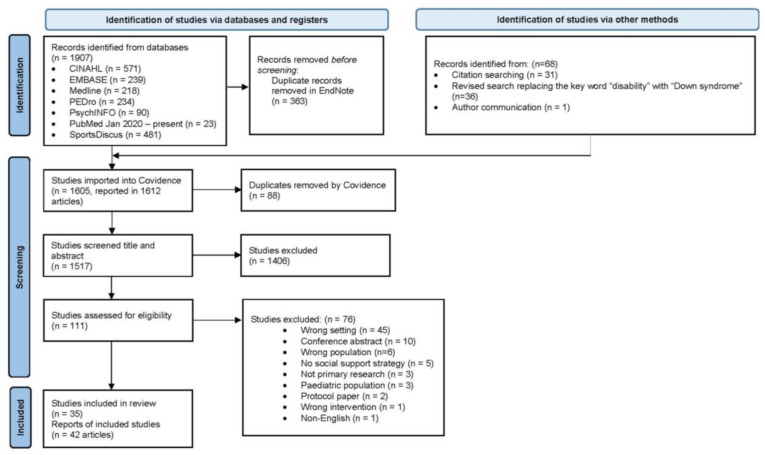
PRISMA flowchart of study selection.

**Figure 2 ijerph-20-00699-f002:**
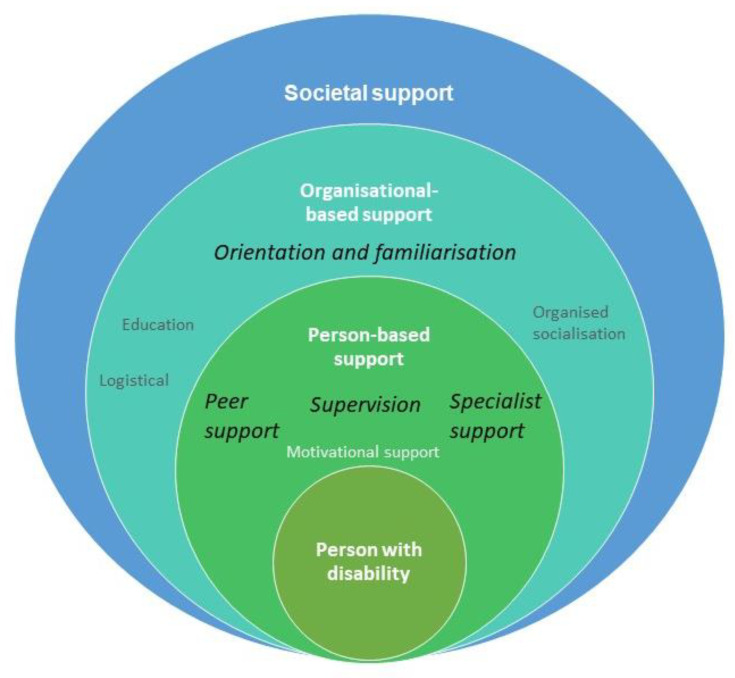
Mapping social supports to the socio-ecological model. The most common person- and organisational-based supports are highlighted in italics.

**Table 1 ijerph-20-00699-t001:** Characteristics of included studies grouped according to whether participants were adolescents and young adults (15–30 years) or adults (30+ years).

**Authors**	**Year**	**Country**	**Participants Condition**	**Sample Size (*n*)**	**Female** **(*n*)**	**Age (yrs)** **Mean (SD) [Range] ***	**Study Design**
**Adolescents and Young Adults (15–30 Years) *n* = 11 Studies**							
Shields et al. [33]	2008	Australia	Down syndrome	20	7	26.8 (7.8)	Randomised controlled trial
Shields & Taylor [17]	2010	Australia	Down syndrome	23	6	15.6 (1.6)	Randomised controlled trial
Shields et al. [18]	2013	Australia	Down syndrome	68	30	17.9 (2.6)	Randomised controlled trial
Temple & Stanish ^#^ [47] Stanish & Temple ^#^ [57]	2011 2012 ^b^	Canada	Intellectual disability	20	10	17.8 (1.6)	Pre-post-test
Stanish & Temple [55]	2012 ^a^	Canada	Intellectual disability	10	5	17.9 (2.3)	Within subject
Pett et al. [38]	2013	USA	Intellectual disability	23	13	24.2 (4.2)	Randomised controlled trial
Shields et al. [32]	2020	Australia	Prader-Willi syndrome	16	8	25.8 (8.2)	Randomised controlled trial
Taylor et al. ^#^ [36] Bania et al. ^#^ [58]	2013 2016	Australia	Cerebral palsy	48	22	18.4 (2.4)	Randomised controlled trial
Zanudin et al. [48]	2021	Malaysia	Cerebral palsy	16	6	19.3 (3.1) [16–24]	Pre-post-test
Shields et al. [53]	2019	Australia	Disability	19	9	18.4 (4.5)	Feasibility study
Shields et al. ^#^ [34] McKenzie et al. ^#^ [15]	2021 2022	Australia	Disability Cerebral Palsy	163 39	61 18	20.8 (5.0) [13–30] 20.4 (4.6)	Stepped wedge RCT Qualitative
**Adults (>30 years) *n* = 24 studies**							
Carter et al. [54]	2004	USA	Developmental disability	15	1	44 [29–69]	Case report
Obrusnikova et al. [56]	2019	USA	Intellectual disability	3	3	[24–37]	Multiple-baseline single-subject
Kovacic et al. [29]	2020	Slovenia	Intellectual disability	150	NR	[18–50]	Randomised controlled trial
Obrusnikova et al. ^#^ [59] Obrusnikova et al. ^#^ [40]	2021 ^a^ 2021 ^b^	USA	Intellectual disability	24	8	26.4 (7.7) [19–44]	Randomised controlled trial
Allen et al. ^#^ [50] Taylor et al. ^#^ [45]	2004 2004	Australia	Cerebral palsy	10 11	4 4	45.8 (5.4) [40–56] 47.6 (8.2) [40–66]	Qualitative Pre-post test
Dodd et al. ^#^ [51] Taylor et al. ^#^ [46]	2006 2006	Australia	Multiple sclerosis	9 9	7 7	45.6 (10.7) [27–61] 45.6 (10.7)	Qualitative Pre-post-test
Dodd et al. [25]	2011	Australia	Multiple sclerosis	71	52	49.1	Randomised controlled trial
Pau et al. [31]	2018	Italy	Multiple sclerosis	22	10	46	Randomised controlled trial
Hassett et al. [14]	2009	Australia	Traumatic brain injury	62	9	34	Randomised controlled trial
Morris et al. [43]	2009	Australia	Traumatic brain injury	7	1	[21–63]	Single-system AB
Hoffman et al. [28]	2010	USA	Traumatic brain injury	80	45	38.4	Randomised controlled trial
Sims et al. [35]	2009	Australia	Stroke	45	18	67.1 (15.2)	Pilot RCT
Handlery et al. [49]	2022	USA	Stroke	15	3	67.6 (11.6)	Pre-post-test
Poliakoff et al. [39]	2013	UK	Parkinson disease	32	11	65.2	Pilot RCT
Corcos et al. [24]	2013	USA	Parkinson disease	48	20	59	Randomised controlled trial
Collett et al. [23]	2016	UK	Parkinson disease	105	44	66.5	Randomised controlled trial
Danoudis & Iansek [41]	2021	Australia	Parkinson disease	17	4	70 (6.7) [59–79]	Pre-post-test
Hansen et al. [52]	2018	Denmark	Glioma	24	10	62 [20–77]	Feasibility study
Fenton et al. [27]	2021	UK	Rheumatic arthritis	70	46	56.4 (12.3)	Randomised controlled trial
Lampousi et al. [30]	2020	Sweden	Mobility disability	110	90	35.1 (6.4)	Randomised controlled trial
Elsworth et al. ^#^ [26] Winward et al. ^#^ [60]	2011 2011	UK	PD, MS, MND, NMD, CP, TBI, TM	99	48	56 (13)	Randomised controlled trial
Ploughman et al. [44]	2014	Canada	Stroke, PD, MS, TBI, NMD	27	10	57.7 (13.6) [32–78]	Repeated measures
Wallace et al. [37]	2019	UK	CMT, IBM	45	14	CMT 46 [39–52] IBM 62 [56–67]	Randomized crossover trial
Duret et al. [42]	2020	France	Stroke, PN, TBI, SCI, other	79	33	59 (14)	Pre-post-test

* Age reported as mean age of all participants; where the mean age of all participants was not reported, the average of the mean ages of each group has been calculated; ^#^ Separate papers reporting the same study; Obrusnikova et al., 2021 ^a^ reports the findings from Stage 1 (first 3-week familiarisation stage), 2021 ^b^ reports the full study Stage 1 and Stage 2 (10-week exercise program).

**Table 2 ijerph-20-00699-t002:** Summary of physical activity interventions, intervention duration and social support strategies.

Authors (Year)	Physical Activity Intervention	Intervention Duration	Supervision	Peer support	Specialist Support	Orientation	Education	Formal Social Activities	Logistical Support	Motivational
Pett et al. (2013) [38]	Aerobic and strength training	12 weeks	X	X					X	
Stanish & Temple (2012 ^b^) ^#^ [57] Temple & Stanish (2011) ^#^ [47]	Aerobic and strength training	15 weeks	X	X		X				
Stanish & Temple (2012 ^a^) [55]	Aerobic and strength training	12 weeks	X	X						
Zanudin et al. (2021) [48]	Aerobic and strength training	18 weeks	X		X	X				
Shields et al. (2019) [53]	Progressive resistance and aerobic training	12 weeks		X	X					
Shields et al. (2021) ^#^ [34] McKenzie et al. (2022) ^#^ [15]	Progressive resistance and aerobic training	12 weeks		X	X	X				
Shields et al. (2020) [32]	Progressive resistance training	10 weeks	X							
Shields et al. (2008) [33]	Progressive resistance training	10 weeks	X	X					X	
Shields & Taylor. (2010) ^+^ [17] Shields et al. (2013) ^+^ [18]	Progressive resistance training	10 weeks		X X	X X					
Taylor et al. (2013) ^#^ [36] Bania et al. (2016) ^#^ [58]	Progressive resistance training	12 weeks	X	X						
Hoffman et al. (2010) [28]	Aerobic training	10 weeks	X		X		X			
Poliakoff et al. (2013) [39]	Aerobic exercise	20 weeks	X	X						
Wallace et al. (2019) [37]	Aerobic training	12 weeks	X		X					
Carter et al. (2004) [54]	Aerobic and strength training	10 weeks	X	X		X			X	
Collett et al. (2016) [23]	Aerobic and strength training	6 months			X	X				
Elsworth et al. (2011) ^#^ [26] Winward et al. (2011) ^#^ [60]	Aerobic and strength training	12 weeks	X		X	X	X		X	
Hansen et al. (2018) [52]	Aerobic and strength training	6 weeks (2nd part)	X		X	X				
Hassett et al. (2009) [14]	Aerobic and strength training	12 weeks	X		X					X
Lampousi et al. (2020) [30]	Aerobic and strength training	12 weeks	X				X			X
Duret et al. (2020) [42]	Aerobic, strength and balance training	6 months	X	X						
Danoudis & Iansek (2021) [41]	Aerobic, progressive resistance and balance training	12 months	X	X	X	X				
Handlery et al. (2022) [49]	Aerobic and strength training alternating with other physical activities	Part 1–8 weeks Part 2–19 weeks	X	X	X	X	X	X	X	
Kovacic et al. (2020) [29]	Balance training (Group 1) Aerobic and strength training and wellness program (Group 2)	16 weeks	X	X ^Gp 1 only^	X ^Gp 1 only^		X			X
Pau et al. (2018) [31]	Aerobic, gait and strength training	24 weeks	X	X	X					
Ploughman et al. (2014) [44]	Aerobic, strength and balance exercises	10 weeks	X	X	X					
Allen et al. (2004) ^#^ [50] Taylor et al. (2004) ^#^ [45]	Progressive resistance training	10 weeks	X	X		X		X		
Corcos et al. (2013) [24]	Progressive resistance training	24 months	X							X
Dodd et al. (2006) ^#^ [51] Taylor et al. (2006) ^#^ [46]	Progressive resistance training	10 weeks	X	X		X		X		
Dodd et al. (2011) [25]	Progressive resistance training	10 weeks	X	X				X		
Morris et al. (2009) [43]	Progressive resistance training	8 weeks	X			X				
Sims et al. (2009) [35]	Progressive resistance training	10 weeks	X	X					X	
Obrusnikova et al. (2019) ^+^ [56]	Strength training	9 sessions	X		X	X	X			
Obrusnikova et al. (2021 ^a^) ^+ # ^^ [59] Obrusnikova et al. (2021 ^b^) ^# ^^ [40]	Strength training	3 weeks 13 weeks	X		X	X	X			X ^2021b only^
Fenton et al. (2021) [27]	RA-tailored exercise	3 months	X		X	X				

^#^ Separate papers reporting the same study; ^^^ Same study, 2021 ^a^ reports 3-week familiarisation phase, 2021 ^b^ reports full study 3 weeks familiarisation and 10-week intervention study; ^+^ Separate studies with similar methods; RA rheumatoid arthritis. CMT Charcot-Marie-Tooth disease; CP cerebral palsy; IBM inclusion body myositis; MND motor neurone disease; MS multiple sclerosis; NMD neuromuscular disease; NR not reported; PD Parkinson’s Disease; PN peripheral neuropathy; RCT randomised controlled trial; SCI spinal cord injury; TBI traumatic brain injury; TM transverse myelitis; UK United Kingdom; USA United States of America. Where two studies or intervention groups are reported, strategies specific to one study or group will be specified. Where not specified, the strategies are used by both studies or intervention groups.

## Data Availability

No new data was created.

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
