# Peer review of "Social Support Initiatives That Facilitate Exercise Participation in Community Gyms for People with Disability: A Scoping Review"

_ijerph, 2022, doi:10.3390/ijerph20010699_

Round 1

Reviewer 1 Report

1.This manuscript provides useful information and suggestions of social support initiatives that facilitate exercise participation in community gyms forpeople with disability, based on current reference review.

2.  The definition of the scoping reviews in this study should have a detail description in this manuscript.

3.The result difference between the exercise facilities owned by municipal government authority and private company should have a discussion.

4.With regard to the effects of the fee or price of community gyms toward the participation, it should have a discussion. 

Author Response

Reviewer 1

  1. The definition of the scoping reviews in this study should have a detail description in this manuscript.

Response:

We defined scoping reviews in our original manuscript as follows:

Scoping reviews systematically map available literature on a topic [21] and examine the extent (size), range (variety) and nature (characteristics) of the evidence [20] (Page 2, line 79).

We have provided further detail and a description of the reasons for choosing to do a scoping review in our revised manuscript. The text now reads:

Scoping reviews provide an overview of the evidence, systematically mapping the available literature on a topic [21] and examining the extent (size), range (variety) and nature (characteristics) of the evidence [20]. They also enable a broad examination of available evidence to identify knowledge gaps, to clarify key concepts related to the area of interest and to report the types of evidence that may inform health practice and practice change [22]. (Page 2, lines 78-83)

Grimshaw, J. (2010). A guide to knowledge synthesis: a knowledge synthesis chapter. Canadian Institutes of Health Research. https://cihr-irsc.gc.ca/e/41382.html

JBI Manual for Evidence Synthesis. (2020). (M. Z. Aromataris E, Ed.). JBI. https://doi.org/https://doi.org/10.46658/JBIMES-20-01

Tricco, A. C., Lillie, E., Zarin, W., O’Brien, K. K., Colquhoun, H., Levac, D., Moher, D., Peters, M. D. J., Horsley, T., Weeks, L., Hempel, S., Akl, E. A., Chang, C., McGowan, J., Stewart, L., Hartling, L., Aldcroft, A., Wilson, M. G., Garritty, C., . . . Straus, S. E. (2018). PRISMA Extension for Scoping Reviews (PRISMA-ScR): Checklist and Explanation. Annals of Internal Medicine, 169(7), 467-473. https://doi.org/10.7326/m18-0850 %m 30178033

3.The result difference between the exercise facilities owned by municipal government authority and private company should have a discussion.

Response:

Thank you for this suggestion. We agree that the way social support is provided in community gyms compared to privately owned and operated gyms might differ. We have expanded the section on limitations in our discussion to address this. The text now reads:

There are limitations in this review relating to the facilities, settings and infrastructure to which the findings primarily relate (community gyms). The types of facilities in this review are more likely to be available in developed countries and may be governed by different policies and funding sources compared with privately owned gyms. Local government facilities usually exist to provide access to recreation for the whole community, including those with disability, and therefore implementing social supports within these settings would align with health and inclusion policies. This scoping review is the first of a three-phase project investigating how social supports are implemented in current practice in community gyms and the perspectives and lived experience of this type of support among young adults with disability who attend community gyms (Kennedy, 2021). The principles underpinning many of the social support strategies identified in this review may also have application beyond community gym settings. (Page 15, line 225-236)

4.With regard to the effects of the fee or price of community gyms toward the participation, it should have a discussion

Response:

We discussed cost in our original manuscript and have provided further detail in the revised manuscript as follows:

One-to-one support is resource intensive and differs from the common practice in gym settings of indirect supervision where a gym trainer oversees all members exercising in the gym space. The provision of social support can facilitate participation in gym settings, but it is important to consider how such support can be implemented outside of funded trials. Disability funding initiatives, such as the National Disability Insurance Scheme in Australia (NDIS https://www.ndis.gov.au/), may cover the costs of 1:1 support in the gym, from a support worker, personal trainer or specialist (e.g., physiotherapist or exercise physiologist). Collaboration between gym facilities, local government, funding bodies (e.g., NDIS) and available supports (e.g., specialists and carers) provide an opportunity to maximise the resources and skills of each sector to facilitate the participation of people with disability in community gym settings. (Pages 13-14, lines 127-137)

And,

However, the cost of entry fees or gym membership is often not covered. In some instances, philanthropic and not-for-profit organisations within the gym and recreation industry offer reduced and low or no-cost membership options for disadvantaged groups, including people with disability. These programs, designed to bridge the equity gap and enable access and inclusion for all to gym and recreation facilities are not, however, widespread across the industry. (Page 15, lines 195-200)

And,

Given the physical, mental and social connection benefits of physical activity participation for people with disability [62,63], there is a need for economic evaluation from a societal perspective of social support initiatives within gym and recreation facilities to inform policy. (Page 15, lines 201-204)

Shields, N., & Synnot, A. (2016). Perceived barriers and facilitators to participation in physical activity for children with disability: a qualitative study. BMC Pediatrics, 16, 9. https://doi.org/10.1186/s12887-016-0544-7

Starowicz, J., Pratt, K., McMorris, C., & Brunton, L. (2022). Mental Health Benefits of Physical Activity in Youth with Cerebral Palsy: A Scoping Review. Physical & Occupational Therapy in Pediatrics, 42(4), 434-450. https://doi.org/10.1080/01942638.2022.2060058

Verschuren, O., Peterson, M. D., Balemans, A. C., & Hurvitz, E. A. (2016). Exercise and physical activity recommendations for people with cerebral palsy. Developmental Medicine & Child Neurology, 58(8), 798-808. https://doi.org/10.1111/dmcn.13053

Reviewer 2 Report

The study aims to look at how social support has been implemented for people with disabilities in research studies in community gyms. 

I'm afraid I do not understand the relevance of this review in the context of enabling those with a disability to access social support in community gyms. It would seem a more relevant question to understand what social support is currently in place, or what the percieved barriers are currently in community gyms that prevents disabled people using them. Looking at previous research studies, which include those that have implemented exercise interventions, does not seem like it answers how gyms can better provide social support, since the provision and support will have been manipulated for each research study.

Perhaps a review that brings together people's current views on gyms or evaluates community based projects cuold be a better criteria, rather than RCT and similar designs that manipulate exercise and support given to disabled individuals.   

Author Response

Reviewer 2

I'm afraid I do not understand the relevance of this review in the context of enabling those with a disability to access social support in community gyms. It would seem a more relevant question to understand what social support is currently in place, or what the perceived barriers are currently in community gyms that prevents disabled people using them. Looking at previous research studies, which include those that have implemented exercise interventions, does not seem like it answers how gyms can better provide social support, since the provision and support will have been manipulated for each research study.

Response:

We have added to the following information to the final paragraph of the discussion to clarify the relevance of the scoping review.

This scoping review is the first of a three-phase project investigating how social supports are implemented in current practice in community gyms and the perspectives and lived experience of this type of support among young adults with disability who attend community gyms (Kennedy, 2021). (Page 16, lines 231-235)

We believe reviewing the literature to identify previous research studies that have implemented complex exercise interventions that include social support strategies can help inform how gyms can better provide social support to those with disability. This type of review can (1) provide examples of what is feasible within community gym settings, and (2) the included studies comprise the evidence-base of what is effective. The review also provides examples of what programs have been implemented in community practice (for example, Shields et al., 2021). We also believe reviewing the literature is an important starting point because a research-to-practice gap may exist; that is, current practice may not be evidence-based.  

Kennedy, R. A., Shields, N. (2021, 19/12/2022). Social support to facilitate physical activity in community gyms for young adults with disability: A scoping review protocol. Center for Open Science.

Shields, N., Willis, C., Imms, C., McKenzie, G., van Dorsselaer, B., Bruder, A. M., Kennedy, R. A., Bhowon, Y., Southby, A., Prendergast, L. A., Watts, J. J., & Taylor, N. F. (2021). Feasibility of scaling-up a community-based exercise program for young people with disability. Disability & Rehabilitation, 1-13. https://doi.org/https://dx.doi.org/10.1080/09638288.2021.1903103

Perhaps a review that brings together people's current views on gyms or evaluates community based projects could be a better criteria, rather than RCT and similar designs that manipulate exercise and support given to disabled individuals.   

Response:

We agree that exploring people's current views on gyms and the evaluation of community-based projects is important. This is beyond the scope of this review in which we collated the existing evidence base to understand how social support have previously been provided in community gym contexts for people with disability. We believe it is important to know whether ideas or designs have worked in practice. In the second and third phases of a larger project we will explore people's current views on social support in gyms and describe how social support is implemented in current practice.  

While outside of our scoping review aim, a subset of studies did report data on the efficacy of these social support strategies. Participants reported facets of social support important to them including peer support from someone with or without disability (Allen et al., 2004; Dodd et al., 2006; Poliakoff et al., 2013; Shields et al., 2019; Shields et al., 2021; Temple & Stanish, 2011); supervision  (Danoudis & Iansek, 2021; Hansen et al., 2018; Ploughman et al., 2014; Shields et al., 2019; Wallace et al., 2019; Zanudin et al., 2021); and having a knowledgeable supervisor (physiotherapist or exercise professional) or mentor (Allen et al., 2004; Danoudis & Iansek, 2021; Dodd et al., 2006; Hansen et al., 2018; Ploughman et al., 2014; Poliakoff et al., 2013; Shields et al., 2021; Wallace et al., 2019).

To this end we have revised the discussion as follows;

This is consistent with qualitative studies where participants with disability reported that supervision from a knowledgeable supervisor or mentor was important to them (Allen et al., 2004; Danoudis & Iansek, 2021; Dodd et al., 2006; Hansen et al., 2018; Ploughman et al., 2014; Poliakoff et al., 2013; Shields et al., 2019; Shields et al., 2021; Wallace et al., 2019; Zanudin et al., 2021). (Page 13, lines 124-126)

We have also added citations to the leading sentence of the third paragraph of the discussion.

Peer support, provided by people with and without disability, was usually an adjunct to specialist support and was important to those with disability (Allen et al., 2004; Dodd et al., 2006; Poliakoff et al., 2013; Shields et al., 2019; Shields et al., 2021; Temple & Stanish, 2011). (Page 14, line 143)

Allen, J., Dodd, K. J., Taylor, N. F., McBurney, H., & Larkin, H. (2004). Strength training can be enjoyable and beneficial for adults with cerebral palsy. Disability & Rehabilitation, 26(19), 1121-1127. http://ez.library.latrobe.edu.au/login?url=https://search.ebscohost.com/login.aspx?direct=true&db=s3h&AN=106591345&site=ehost-live&scope=site

Danoudis, M., & Iansek, R. (2021). A long-term community gym program for people with Parkinson's disease: a feasibility study of the Monash Health "Health and Fitness" model. Disability & Rehabilitation, 1-9. https://doi.org/10.1080/09638288.2021.1977396

Dodd, K. J., Taylor, N. F., Denisenko, S., & Prasad, D. (2006). A qualitative analysis of a progressive resistance exercise programme for people with multiple sclerosis. Disability & Rehabilitation, 28(18), 1127-1134. http://ez.library.latrobe.edu.au/login?url=https://search.ebscohost.com/login.aspx?direct=true&db=s3h&AN=106210559&site=ehost-live&scope=site

Hansen, A., Søgaard, K., Minet, L. R., & Jarden, J. O. (2018). A 12-week interdisciplinary rehabilitation trial in patients with gliomas - a feasibility study. Disability & Rehabilitation, 40(12), 1379-1385. http://ez.library.latrobe.edu.au/login?url=https://search.ebscohost.com/login.aspx?direct=true&db=s3h&AN=128422256&site=ehost-live&scope=site

Ploughman, M., Shears, J., Harris, C., Hogan, S. H., Drodge, O., Squires, S., & McCarthy, J. (2014). Effectiveness of a novel community exercise transition program for people with moderate to severe neurological disabilities. Neurorehabilitation, 35(1), 105-112. https://doi.org/https://dx.doi.org/10.3233/NRE-141090

Poliakoff, E., Galpin, A. J., McDonald, K., Kellett, M., Dick, J. P., Hayes, S., & Wearden, A. J. (2013). The effect of gym training on multiple outcomes in Parkinson's disease: a pilot randomised waiting-list controlled trial. Neurorehabilitation, 32(1), 125-134.

Shields, N., & Synnot, A. (2016). Perceived barriers and facilitators to participation in physical activity for children with disability: a qualitative study. BMC Pediatrics, 16, 9. https://doi.org/10.1186/s12887-016-0544-7

Shields, N., van den Bos, R., Buhlert-Smith, K., Prendergast, L., & Taylor, N. (2019). A community-based exercise program to increase participation in physical activities among youth with disability: a feasibility study. Disability & Rehabilitation, 41(10), 1152-1159. https://doi.org/https://dx.doi.org/10.1080/09638288.2017.1422034

Shields, N., Willis, C., Imms, C., McKenzie, G., van Dorsselaer, B., Bruder, A. M., Kennedy, R. A., Bhowon, Y., Southby, A., Prendergast, L. A., Watts, J. J., & Taylor, N. F. (2021). Feasibility of scaling-up a community-based exercise program for young people with disability. Disability & Rehabilitation, 1-13. https://doi.org/https://dx.doi.org/10.1080/09638288.2021.1903103

Temple, V. A., & Stanish, H. I. (2011). The feasibility of using a peer-guided model to enhance participation in community-based physical activity for youth with intellectual disability. Journal of Intellectual Disabilities, 15(3), 209-217.

Wallace, A., Pietrusz, A., Dewar, E., Dudziec, M., Jones, K., Hennis, P., Sterr, A., Baio, G., Machado, P. M., Laurá, M., Skorupinska, I., Skorupinska, M., Butcher, K., Trenell, M., Reilly, M. M., Hanna, M. G., & Ramdharry, G. M. (2019). Community exercise is feasible for neuromuscular diseases and can improve aerobic capacity. Neurology, 92(15), e1773-e1785. https://doi.org/10.1212/wnl.0000000000007265

Zanudin, A., Mercer, T., Samaan, C., Jagadamma, K., McKelvie, G., & van der Linden, M. (2021). A community-based exercise program for ambulant adolescents and young adults with cerebral palsy, a feasibility study. European Journal of Adapted Physical Activity.

Reviewer 3 Report

Given the lows rates of participation of disabled people in active recreation, this scoping review of support initiatives that can facilitate their participation in community gyms is useful.  You have clearly set out your methods and findings and mapped these to a socio-ecological model. All of the studies included in the paper report on researcher-initiated and funded trials - and you state it is unclear how community gyms could implement the social supports identified given resources required to do so. It would be useful to know if there are community gyms which have support programmes in place to support disabled participants, and if so, how they fund and implement these.

Similarly, you effectively map the social supports identified in this scoping review to a socio-ecological model; it would be good to also reflect on the potential impact of wider organisational/policy/environmental (social and physical) dimensions of the socio-ecological model that could facilitate or provide a barrier to social support programmes. Reflecting on your findings within the broader socio-ecological context of community gyms would increase their relevance. 

Author Response

Reviewer 3

All of the studies included in the paper report on researcher-initiated and funded trials - and you state it is unclear how community gyms could implement the social supports identified given resources required to do so. It would be useful to know if there are community gyms which have support programmes in place to support disabled participants, and if so, how they fund and implement these.

Response:

We agree it would be useful to know if there are community gyms which have programs in place to provide social support to participants with disability, and to understand how these programs are funded and implemented. This is beyond the scope of this review, but these data will be collected as part of the broader 3-phase program of work we are completing as outlined in our responses to reviewers 1 and 2.

To acknowledge the reviewer’s comments, we have revised the discussion as follows:

However, the cost of entry fees or gym membership is often not covered. In some instances, philanthropic and not for profit organisations offer reduced, low or no-cost membership options for disadvantaged groups, including people with disability. These programs, designed to bridge the equity gap and enable access and inclusion for all to gym and recreation facilities are not, however widespread across the industry. (Page 15, lines 195-200)

Similarly, you effectively map the social supports identified in this scoping review to a socio-ecological model; it would be good to also reflect on the potential impact of wider organisational/policy/environmental (social and physical) dimensions of the socio-ecological model that could facilitate or provide a barrier to social support programmes. Reflecting on your findings within the broader socio-ecological context of community gyms would increase their relevance.

Response:

Thank you for this feedback. To increase the relevance of our findings, we have provided additional discussion on the potential impact of wider organisational, policy, and social and physical environmental factors that could facilitate or act as a barrier to social support programmes. The revised manuscript now includes the following:

The social supports identified in this review primarily map within the socio-ecological model to person-based supports and organisational-based supports for the individual with disability and their close social networks (Figure 2). However, no studies included in this review considered the potential impact of wider organisational, physical and social environmental or policy contexts that could facilitate or hinder the implementation of social support strategies within community gym settings. Local champions may drive practice in the provision of social supports for people with disability within individual community facilities. However, the support of decision-makers within recreation organisations and local government ultimately influences the implementation of inclusive policies and importantly, funding to support these policies. The findings of this review provide a basis for how recreation gym facilities could implement social support to increase participation among people with disability. It also provides exemplars of strategies or programs that could be supported by local government policy and funding.  (Page 15, lines 205-217)